# Analysis on the Temporal Distribution Characteristics of Air Pollution and Its Impact on Human Health under the Noticeable Variation of Residents’ Travel Behavior: A Case of Guangzhou, China

**DOI:** 10.3390/ijerph17144947

**Published:** 2020-07-09

**Authors:** Xiaoxia Wang, Chao Zou, Luqi Wang

**Affiliations:** School of Civil and Transportation Engineering, Guangdong University of Technology, Guangzhou 510006, China; wxx@gdut.edu.cn (X.W.); chao.zou@gdut.edu.cn (C.Z.)

**Keywords:** air pollution, human activity, temporal distribution, particulate matter, gaseous contaminants, travel behavior

## Abstract

During the large-scale outbreak of COVID-19 in China, the Chinese government adopted multiple measures to prevent the epidemic. The consequence was that a sudden variation in residents’ travel behavior took place. In order to better evaluate the temporal distribution of air pollution, and to effectively explore the influence of human activities on air quality, especially under the special situation, this study was conducted based on the real data from a case city in China from this new perspective. Two case scenarios were constructed, in which the research before the changes of residents’ travel behavior was taken as case one, and the research after the changes in residents’ travel behavior as case two. The hourly real-time concentrations of PM_2.5_, PM_10_, SO_2_, NO_2_, CO and O_3_ that have passed the augmented Dickey–Fuller (ADF) test were employed as a data source. A series of detailed studies have been carried out using the correlation method, entropy weight method and the Air Quality Index (AQI) calculation method. Additionally, the research found that the decrease rate of NO_2_ concentration is 61.05%, and the decrease rate of PM_10_ concentration is 53.68%. On the contrary, the average concentration of O_3_ has increased significantly, and its growth rate has reached to 9.82%. Although the air quality in the first week with fewer travels was in the excellent category, and chief pollutant (CP), as well as excessive pollutant (EP), were not found, as traffic volume increased, it became worse in the second and third weeks. In addition to that, special attention should still be paid to the development trend of O_3_, as its average hourly concentration has increased. The findings of this study will have some guiding significance for the study of air pollution prevention, cleaner production, and indoor environmental safety issues, especially for the study of abnormal traffic environments where residents’ travel behaviors have changed significantly.

## 1. Introduction

Outdoor and indoor air pollution poses a serious threat to the sustainable development of society and the economy, thus causing a broad concern for public health and cleaner production [1,2]. Plentiful pieces of literature show that air pollutants including particulate matter, and gaseous contaminants can endanger human health directly or indirectly [3,4,5,6,7,8]. Sulfur dioxide (SO_2_), as a common pollutant, is greatly harmful to living beings [9]. Nitrogen oxides (NO_X_) and particulate matter have known harmful impacts on human health, such as causing damage to the respiratory and cardiovascular systems [10,11,12,13]. Carbon monoxide (CO) is a colorless, tasteless and odorless gas that is poisonous to the human body [14,15]. Ground- and tropospheric-level O_3_ have been shown to have adverse effects on public health, cause respiratory diseases and increase mortality by impairing lung function [16,17,18].

With the rapid development of industrialization and urbanization, atmospheric environmental pollution has become increasingly serious in China in recent decades [19]. To measure air quality in a simple and intuitive way, while letting the public know the real-time status of air quality, the Air Quality Index (AQI) has been adopted by the Chinese Ministry of Environmental Protection (MEP) since 2012. AQI was proposed by the United States Environmental Protection Agency and has been widely used around the world. AQI is a dimensionless index that quantitatively describes the condition of air quality. The larger the AQI is, the more serious the pollution becomes, and the more obvious the impact of air pollution on human health. Research on the impact of atmospheric pollutants on human health based on AQI and the spatial and temporal distribution of pollutants has attracted the attention of many scholars [20,21].

Air pollution is not only related to natural phenomena such as seasons [22,23,24] and wind speeds [25], but also to social phenomena such as human activities [26,27,28]. In December 2019, an “unknown cause of pneumonia” appeared in Wuhan, China. On 7 January 2020, it was confirmed as “novel coronavirus” by whole genome sequencing. On 12 January 2020, the World Health Organization (WHO) temporarily named it “2019-nCoV” (2019 novel coronavirus). On 11 February 2020, WHO officially named it “COVID-19”, where “CO” stands for “Corona”, “VI” stands for “Virus”, and “D” stands for “Disease”. Common signs of a person infected with COVID-19 include respiratory symptoms, fever, cough, shortness of breath, and dyspnea. In more severe cases, the infection can cause pneumonia, severe acute respiratory syndrome, kidney failure, and even death. COVID-19 is highly contagious. As of 21 February 2020, a total of 75,567 confirmed infections and 2239 deaths from infection were reported in China. In order to better prevent and control COVID-19, a series of measures have been introduced to effectively control the spread of COVID-19 in China, such as wearing masks, quarantining people who have returned from affected areas, putting an end to family gatherings, proposing staying at home and not driving around, and forcing factories and schools to shut down. The result of appropriate government policies is that, during the COVID-19 epidemic prevention and control period, human activities, especially the travel behavior of residents, have changed dramatically. During this period, the number of residents traveling by cars and public transit dropped drastically. Instead, residents took close walking or stayed at home without traveling. Correspondingly, the vehicular exhaust emissions have changed, the air pollution has changed, and the air quality has changed. 

In order to better evaluate the temporal distribution of air pollution under the noticeable variation of resident travel behaviors, and to effectively explore the influence of human travel activities on air pollution, such as under the above mentioned abnormal human activities which were caused by COVID-19, this study was carried out from this new perspective, and based on real data from a case city in China. Two case scenarios were constructed to complete the comparison of air pollution and air quality before and after noticeable variation of residents’ travel behavior. The correlation relationship between AQI and six representative pollutants was analyzed, the temporal distribution characteristics of pollutant concentration of six representative pollutants in both cases were carried out, the comparison of entropy weights of their pollutant concentrations in both cases was conducted, and the impact of air pollution on human health was evaluated based on the AQI calculation formula. 

## 2. Data Source and Datasets Stationarity Test

### 2.1. Data Source

Guangzhou, located at (112°57′ E~114°3′ E, 22°26′ N~23°56′ N), is the greatest city and capital city of Guangdong Province in China. It is the third-largest city in China and the largest city in southern China. By the end of 2018, the city had 11 districts, with a total area of 7434.4 square kilometers and a resident population of 13.501 million people. The largest population density of the jurisdiction (Yuexiu District) is 34,225 people per square kilometer. The average population density in Guangzhou is 8582 people per square kilometer. The climate here is a typical monsoon marine climate of the South Subtropics. The annual average temperature is 20°~22°, the average wind speed is 1.6 m/s, and the average relative humidity is 77% [29]. The low average annual wind speed and high population density are exacerbating air pollution. The solution to air pollution in Guangzhou has a strong guiding significance for air pollution control in Guangdong, China, and even for other densely populated countries.

In order to comprehensively monitor air quality, air quality monitoring stations have been established at eleven different locations in Guangzhou, including Sports West station and Luhu Lake station, and so on. The air quality monitoring stations are the basic platform for Guangzhou to collect air pollution concentration data and evaluate air quality. Multi-parameter automatic monitoring instruments are installed in the air quality monitoring station to the conduct fixed-site, continuous and automatic sampling of atmospheric pollutants. Therefore, the real-time concentration of PM_2.5_, PM_10_, SO_2_, NO_2_, CO and O_3_ can be automatically recorded every hour. However, this is affected by many factors such as the geographic location of monitoring stations, the number of nearby residents, and the amount of road traffic; the concentration of PM_2.5_, PM_10_, SO_2_, NO_2_, CO and O_3_ detected by each monitoring station will vary. To fully reflect the overall air quality and the temporal distribution characteristics of pollutants before and after variations in residents’ travel behavior in Guangzhou, the weighted average value of the PM_2.5_, PM_10_, SO_2_, NO_2_, CO and O_3_ concentrations detected by the above mentioned eleven monitoring stations are adopted. The weighted average values are 0.094801, 0.103976, 0.073394, 0.110092, 0.097589, 0.107034, 0.061162, 0.11315, 0.100917, 0.06422 and 0.073394, respectively, at Sports West, Luhu Lake, Jiulong Town Zhenlong, Guangzhou No. 5 Middle School, Guangzhou 86th Middle School, Guangzhou Monitoring Station, Maofeng Mountain Forest Park, Guangdong Business School, Guangya Middle School, Panyu Middle School and Huadu District Normal School station, which are calculated based on the air quality of each monitoring station. Additionally, the hourly real-time concentration of PM_2.5_, PM_10_, SO_2_, NO_2_, CO and O_3_ from 1 January 2020 to 15 February 2020 are contained in this study. These data are sourced from http://beijingair.sinaapp.com/.

In order to better compare the changes in pollutant concentrations before and after variations of residents’ travel characteristics, the hourly concentration of PM_2.5_, PM_10_, SO_2_, NO_2_, CO and O_3_ from 1 January to 23 January 2020 were used as datasets before travel behavior changes. As the notice calling for less travel was issued by the Guangdong provincial government on 23 January 2020, the hourly concentration of PM_2.5_, PM_10_, SO_2_, NO_2_, CO and O_3_ from 24 January to 15 February 2020 as datasets after changes in travel behavior. In the follow-up research, to make the expression more concise and clearer, the study on the change of pollutant concentration before the variation of resident travel behavior is taken as case one. Additionally, the study on the change of pollutant concentration after the variation of resident travel behavior is taken as case two.

### 2.2. Datasets Stationarity Test

Since the hourly real-time concentrations of PM_2.5_, PM_10_, SO_2_, NO_2_, CO and O_3_ from 1 January 2020 to 15 February 2020 are the data observed at different times, they cannot be regarded as being generated by a same random variable. Moreover, to avoid nonstationary of datasets and sensitiveness to departures from stationarity of the presented methodology mentioned below, the above datasets should be regarded as being generated by a random process and their stationary test should be done. As the unit root test is the most common and effective test method in the stationarity test, which includes the Dickey–Fuller (DF) test, the augmented Dickey–Fuller (ADF) test, the Phillips–Perron (PP) test, etc. [30], this section will apply the ADF test, which is widely used to test the time-series data stationarity [30,31], to complete the stationarity test of datasets used in this article. Hence, the time series data of hourly real-time pollutant concentrations ($Y_{t}$) was deemed as a random variable that changes over time (t) and $\left\{Y_{t},t\in T\right\}$ was defined as a random process. If $Y_{t}$ meets the requirements of the Equation (1), then the random process $\left\{Y_{t},t\in T\right\}$ is a basic unit root process.
(1)$$Y_{t}=\eta +Y_{t-1}+\varepsilon_{t}$$
where, $\varepsilon_{t}$ refers to a correction term subject to the white noise process. η is a coefficient.

When ADF test is applied to the test of the random process $\left\{Y_{t},t\in T\right\}$, the basic unit root process could be expanded into the following three versions, as described below.

The model without constants and trends:(2)$$\Delta Y_{t}=\omega Y_{t-1}+\sum_{i=1}^{k}\beta_{i}\Delta Y_{t-i}+\varepsilon_{t}$$

The model with constants and without trends:(3)$$\Delta Y_{t}=\alpha_{0}+\omega Y_{t-1}+\sum_{i=1}^{k}\beta_{i}\Delta Y_{t-i+1}+\varepsilon_{t}$$

The model with constants and trends:(4)$$\Delta Y_{t}=\alpha_{0}+\alpha_{1}t+\omega Y_{t-1}+\sum_{i=1}^{k}\beta_{i}\Delta Y_{t-i+1}+\varepsilon_{t}$$
where, Δ is the 1st difference,$\alpha_{0}$ refers to a constant, $\alpha_{1}t$ represents the trend, ω means the autoregressive term, $\sum_{i=1}^{k}\beta_{i}\Delta Y_{t-i+1}$ stands for the lag of dependent variables, $\varepsilon_{t}$ means the white noise, k is the orders of lags.

The null hypothesis of the ADF unit root test can be stated as *H*_0_: ω = 0, which means that the unit root exists. If the null hypothesis (*H*_0_) is rejected, then the variables with a unit root can be rejected and the datasets can be considered stationary.

After the calculation using the abovementioned ADF test methodology, the ADF test results for PM_2.5_, PM_10_, SO_2_, NO_2_, CO and O_3_ can be concluded as:(1)For the original time-series datasets of PM_2.5_, PM_10_ and SO_2_ concentration, the null hypothesis was rejected, which means that their data series were stationary and there was no unit root. Therefore, these datasets passed the ADF test.(2)For the original time-series datasets of CO and O_3_ concentration, the null hypothesis was rejected, and these datasets passed the ADF test at a 95% confidence level. Therefore, these datasets were stationary at a 95% confidence level.(3)For the original time-series data of NO_2_ concentration, the null hypothesis was accepted (which can be seen from the ADF test results before the data transformation in Table 1), so there existed a unit root. Therefore, these datasets were nonstationary.

To ensure that the effects of time, trends, etc. are eliminated before applying the datasets to subsequent analysis, and to avoid time-series data being nonstationary, the transformation of the original NO_2_ datasets was carried out using the exponentially weighted moving average (EWMA) method [32], which combines the log transform method and moving average method together. The ADF test of the processed data after EWMA transformation can be seen from the results in Table 1. Meanwhile, the log-transformed, moving average and the weighted moving average of NO_2_ datasets can be seen in Figure 1. The NO_2_ concentration datasets after log-transformed, and their rolling mean as well as their rolling standard deviation, can be seen in Figure 2. Judging from the results after the data transformation in Table 1 (the p-value is significantly less than 0.05) and the rolling standard deviation in Figure 2 (the rolling standard deviation fluctuates around 0), the conclusion that the processed data after EWMA transformation can completely reject the null hypothesis and pass the ADF test can be drawn. Therefore, the transformed NO_2_ concentration datasets were obtained through log-recover and roll-recover.

From now on, the original time-series datasets of PM_2.5_, PM_10_, SO_2_, CO and O_3_, as well as the transformed time-series datasets of NO_2_ concentration, were used for the subsequent analysis, as these datasets passed the data stationary of the ADF test.

## 3. Temporal Distribution of Six Representative Pollutions

### 3.1. The Correlation Relationship between AQI and Six Representative Pollutants 

The value of AQI is calculated by comparing the concentration of a pollutant to that of a series of air pollutants, and the AQI on a specific hour or specific day is determined by the pollutant(s) that have the highest AQI score among all representative air pollutants [4,21,33]. It can be seen that the level of AQI is directly related to the concentration of pollutants, but the impact of each pollutant concentration on AQI is different. To reflect the influence of each pollutant concentration on AQI, the correlation coefficients between AQI and six representative pollutants in both cases are calculated according to Equation (5). Additionally, the correlation relationships (ρ) between AQI and their concentrations in both cases are shown in Figure 3 and Figure 4, respectively. It can be seen intuitively from Figure 3 and Figure 4, no matter in case one or case two, the correlation between AQI and particulate matter is obvious, but its correlation with gaseous contaminants is quite complex. Subsequently, the calculation results are shown in Table 2.
(5)$$r(x,y)=\frac{\mathrm{Cov}(x,y)}{\sqrt{\mathrm{Var}[x]\mathrm{Var}[y]}}$$
where, r(x,y) refers to correlation coefficients between variable *x* and *y*, Cov(x,y) is covariance between variable *x* and *y*, Var[x] is variance of variable *x*, Var[y] is variance of variable *y*.
(6)$${r(\mathrm{AQI},\;{\mathrm{PM}}_{2.5})}_{\mathrm{case}\;\mathrm{one}}=0.939462,\;{r\;(\mathrm{AQI},\;{\mathrm{PM}}_{10})}_{\mathrm{case}\;\mathrm{one}}=0.978872$$
(7)$${r(\mathrm{AQI},\;{\mathrm{PM}}_{2.5})}_{\mathrm{case}\;\mathrm{two}}=0.994004,\;{r\;(\mathrm{AQI},\;{\mathrm{PM}}_{10})}_{\mathrm{case}\;\mathrm{two}}=0.986916$$

The above Equations (6) and (7) indicate that the relationship between AQI and particulate matter are highly correlated.
(8)$${r(\mathrm{AQI},\;{\mathrm{SO}}_{2})}_{\mathrm{case}\;\mathrm{one}}=0.707992,\;{r\;(\mathrm{AQI},\;{\mathrm{NO}}_{2})}_{\mathrm{case}\;\mathrm{one}}=0.796998$$

While,
(9)$${r(\mathrm{AQI},\;{\mathrm{SO}}_{2})}_{\mathrm{case}\;\mathrm{two}}=0.491164,\;{r\;(\mathrm{AQI},\;{\mathrm{NO}}_{2})}_{\mathrm{case}\;\mathrm{two}}=0.601048$$

In the scenario of case one, the relationship between AQI and SO_2_, NO_2_ are moderately correlated. In the scenario of case two, AQI and NO_2_ are still moderately correlated. However, the correlation between AQI and SO_2_ is significantly reduced, and even the relationship between AQI and SO_2_ belongs to lowly correlated. Medium and low correlations do not indicate that the impact of SO_2_ and NO_2_ on AQI is decreasing, but merely that the relationship between them is not very distinctive in terms of linear correlation, and that the relationship between them is more complicated. The relationship between them will be explained in the following chapters.
(10)$${r(\mathrm{AQI},\;\mathrm{CO})}_{\mathrm{case}\;\mathrm{one}}=0.125977,\;{r\;(\mathrm{AQI},\;O_{3})}_{\mathrm{case}\;\mathrm{one}}=0.018185$$
(11)$${r(\mathrm{AQI},\;\mathrm{CO})}_{\mathrm{case}\;\mathrm{two}}=0.130621,\;{r\;(\mathrm{AQI},\;O_{3})}_{\mathrm{case}\;\mathrm{two}}=0.053570$$

The relationship between AQI and CO, O_3_ is extremely weak in both cases. The discrepancy in the coefficient of O_3_ in case two through comparing to case one indicates that O_3_ has increased. At the same time, it also indicates that the concentrations of O_3_ have risen after the variation of residents’ travel behaviors.

The average concentration of pollutants in the two cases has changed significantly. As shown in Table 3, the average concentration of PM_2.5_ decreased from 32.28 ${\mu g/m}^{3}$ to 21.77 ${\mu g/m}^{3}$. The average concentration of PM_10_ decreased from 61.38 ${\mu g/m}^{3}$ to 28.43 ${\mu g/m}^{3}$. The average concentration of SO_2_ decreased from 6.86 ${\mu g/m}^{3}$ to 4.99 ${\mu g/m}^{3}$. The average concentration of NO_2_ decreased from 52.22 ${\mu g/m}^{3}$ to 20.34 ${\mu g/m}^{3}$. The average concentration of CO decreased from 0.98 ${\mathrm{mg}/m}^{3}$ to 0.74 ${\mathrm{mg}/m}^{3}$. However, the average concentration of O_3_ increased from 40.14 ${\mu g/m}^{3}$ to 44.08 ${\mu g/m}^{3}$. Table 3 also shows the decline rates of the average concentrations of six representative pollutants in case two, by comparison with case one. Among them, the decrease rate of NO_2_ and PM_10_ are the most considerable. The decrease rate of NO_2_ concentration is 61.05%, and the decrease rate of PM_10_ concentration is 53.68%. On the contrary, the concentration of O_3_ has increased significantly, and the growth rate of its average concentration has reached to 9.82%. 

### 3.2. Pollutant Concentration Diurnal Characteristics 

The temporal distribution characteristics of six representative pollutant concentrations in both cases over 24-h a day are shown in Figure 5 and Figure 6, respectively.

PM_2.5_ and PM_10_ are usually the chief pollutants in Guangzhou urban air pollution. The changing tendency of PM_2.5_ and PM_10_ is almost the same, both in case one and case two. PM_2.5_ and PM_10_ concentration have the characteristic of bimodal distribution in diurnal variation. Their concentration peaks at around 10:00 and 23:00 in case one, and at around 11:00 and 22:00 in case two. Through the above comparative analysis, it can be obtained that the variation in residents’ travel behavior has brought down the concentrations of PM_2.5_ and PM_10_, and it has not changed the characteristics of the bimodal distribution of their concentrations. However, it will cause their concentration peaks to appear about an hour later in the morning and about an hour earlier in the evening. This diurnal characteristic is similar to the characteristics of residents’ travel activities. Significant changes in residents’ travel characteristics during the COVID-19 epidemic have led to a significant reduction in the total amount of urban travel demand and a sudden disappear of normalized urban road traffic congestion. Not only is the road space resource extremely surplus, but also the peak time of traffic flow is delayed. This is in accordance with the characteristics of the time-varying law of pollutants. SO_2_ has the characteristic of approximately unimodal distribution and its concentration peaks at around 12:00 in case one. However, the general trend of its concentration was relatively flat and there was no significant fluctuation in case two, except for a few abnormal values that are measured from 1:00 to 4:00 in the morning. Through the above comparative analysis, it can be obtained that the variation in residents’ travel behavior has brought down the concentrations of SO_2_, and it is one of the important factors in determining whether the peak of SO_2_ concentration occurs. Although SO_2_ pollution is directly related to industrial pollution sources [34], it also has a positive correlation with the traffic flow [35]. During the COVID-19 epidemic, the amount and the time-varying characteristics of traffic flow are different from the past. For this reason, the concentration of SO_2_ pollutants also showed corresponding changes. The O_3_ concentration peaked at around 17:00 in the afternoon, and the NO_2_ concentration was approximately U-shaped. The NO_2_ concentration rises rapidly at around 18:00 every day, and then slowly drops after reaching the peak of pollution at 21:00 in case one. Compared with case one, the O_3_ concentration fluctuates greatly, with a smaller peak at 6:00 in the morning and a bigger peak at 17:00 in the afternoon in case two. Additionally, the NO_2_ concentration has the characteristic of U-shaped distribution, which rises rapidly at around 18:00 every day, and then slowly drops after reaching the peak of pollution at around 23:00 at night. Through the above comparative analysis, it can be obtained that residents’ travel behavior will change the concentration distribution of O_3_ from unimodal to bimodal. Additionally, it will cause the peak concentration of NO_2_ to appear two hours earlier in the evening. The significant increase of NO_2_ in the evening period may be due to the superposition of two reasons. The first is the concentrated emission of vehicular exhaust during peak hours at night, and the second is the accumulation of NO_2_ pollution at night after the ozone pollution subsides during the daytime.

The change of CO concentration with time in case one is stable, while the change in case two is disordered. Through the above comparative analysis, it can be obtained that the variation in residents’ travel behavior did not bring about a simultaneous change in CO concentration.

## 4. The Entropy Weight of Six Pollutants

The third chapter mainly studies the correlation relationship between six pollutants and AQI, and the temporal distribution characteristics of their respective concentrations. However, the contribution of each pollutant to overall air quality has not been studied. Next, the entropy weight method was adopted to mine the contribution. 

The entropy weight method is one of the classic algorithms for calculating indicator weights [36,37,38]. It was initially derived from Shannon entropy. In 1948, Shannon introduced the concept of entropy into information theory, as a measure of information uncertainty based on probability theory [39]. Nowadays, the Shannon entropy has been widely used in engineering technology, social economy and other fields [40,41]. The Shannon entropy is an objectively assigned weight method. The biggest difference between it and the subjective weight distribution method, such as the analytic hierarchy process method (AHP), expert survey method, etc., is that it determines the weight of the indicators based on the inherent information of the indicators, which can eliminate human interference and make the result more consistent with the fact [37]. Moreover, the Shannon entropy has a good capability in assessing the time-varying degree of informational efficiency of time-series data [42,43]. According to the characteristics of the Shannon entropy, we can judge the randomness and disorder of an event by calculating the entropy weight. We can also use the entropy weight to determine the degree of discreteness of an indicator. The larger the entropy, the more chaotic the system is, and the less information it carries. The smaller the entropy, the more orderly the system is, and the more information it carries. The calculation procedure of the entropy weight method is described as follows,

(1) Build a data matrix. Assuming that the data have *n* rows of records and *m* feature columns, then the data can be represented by an *n***m* matrix A.
(12)$$A=\left[\begin{array}{cc} \begin{array}{ccc} x_{11} & x_{12} & \cdots \end{array} & x_{1m}\\ \begin{array}{ccc} \begin{array}{c} x_{21}\\ \vdots \\ x_{n1} \end{array} & \begin{array}{c} x_{22}\\ \vdots \\ x_{n2} \end{array} & \begin{array}{c} \cdots \\ \vdots \\ \cdots \end{array} \end{array} & \begin{array}{c} x_{2m}\\ \vdots \\ x_{\mathrm{nm}} \end{array} \end{array}\right]$$

(2) Normalization of indicators. Because the measurement units of the indicators are not uniform, before using them to calculate comprehensive indicators, they must be standardized. That is, the absolute value of the indicator should be converted into a relative value, thereby solving the problem of homogeneity of various qualitative indicator values. Moreover, because the values of the positive and negative indicators represent different meanings (the higher the value of the positive indicator, the better; on the contrary, the lower the value of the negative indicator, the better). Therefore, we can use different algorithms for data normalization for positive and negative indicators. The specific method is as follows:

For positive indicators,
(13)$$x_{\mathrm{ij}}=\frac{x_{\mathrm{ij}}-\min\left\{x_{1j}{,\ldots ,x}_{\mathrm{nj}}\right\}}{\max\left\{x_{1j}{,\ldots ,x}_{\mathrm{nj}}\right\}-\min\left\{x_{1j}{,\ldots ,x}_{\mathrm{nj}}\right\}}\;(i=1,\cdots ,n;j=1,\cdots ,m)$$

For negative indicators,
(14)$$x_{\mathrm{ij}}=\frac{\max\left\{x_{1j}{,\ldots ,x}_{\mathrm{nj}}\right\}{-x}_{\mathrm{ij}}}{\max\left\{x_{1j}{,\ldots ,x}_{\mathrm{nj}}\right\}-\min\left\{x_{1j}{,\ldots ,x}_{\mathrm{nj}}\right\}}\;(i=1,\cdots ,n;j=1,\cdots ,m)$$

(3) Calculate the proportion of the *i*-th record under the *j*-th indicator.
(15)$$p_{\mathrm{ij}}=\frac{x_{\mathrm{ij}}}{\sum_{i=1}^{n}x_{\mathrm{ij}}}\;(i=1,\cdots ,n;j=1,\cdots ,m)$$

(4) Calculate the entropy weight of the *j*-th indicator.
(16)$$e_{j}=-k*\sum_{1}^{n}p_{\mathrm{ij}}{*\log(p}_{\mathrm{ij}}),\;k=1/\ln(n)$$

(5) Calculate the coefficient of variance for the *j*-th indicator.
(17)$$d_{j}{=1-e}_{j}$$

(6) Calculate the weight of the *j*-th indicator.
(18)$$w_{j}=\frac{d_{j}}{\sum_{1}^{m}d_{j}}$$

A comparison of entropy weights of six pollutant concentrations in both cases is reflected in Table 4. Upward or downward arrows are employed to indicate the movement of entropy weights. SO_2_, NO_2_, and PM_2.5_ have got upward arrows. It can be perceived that the dispersion of the hourly measured concentration of SO_2_, NO_2_, and PM_2.5_ has increased. Relatively speaking, the dispersion of the other three pollutants, including CO, PM_10_ and O_3_, is declining. Here, taking SO_2_ and O_3_, which have obvious discrete data values in Table 4 as an example, to visually observe the fluctuation of data over time. The concentration of SO_2_ and O_3_ in both cases are shown in Figure 7 and Figure 8. By comparing the SO_2_ concentration of case one and case two in Figure 7, it can be found that there is a certain degree of data dispersion in the hourly SO_2_ concentration in case one. However, this kind of data dispersion is not as strong as that in case two, especially when abnormal discrete values from 1:00 to 4:00 AM and 16:00 PM were discovered. These abnormal discrete values result in a greater entropy weight for SO_2_ when evaluating the atmospheric quality. Since the average concentrations of SO_2_ in both cases are 6.76 ${\mu g/m}^{3}$ and 5.01 ${\mu g/m}^{3}$, respectively, and the entropy weight of SO_2_ is in an upward trend in case two, it can be considered that the change in residents’ travel behavior will bring about an effective reduction in SO_2_ concentration. Meanwhile, the time-sensitivity of SO_2_ concentration in case two is stronger than that in case one. O_3_ concentration changes diametrically opposite to SO_2_ concentration, as shown in Figure 8. The abnormal discrete values of O_3_ concentration were found from 13:00 to 19:00 PM in case one. Since the average concentrations of O_3_ in both cases are 42.10 ${\mu g/m}^{3}$ and 44.55 ${\mu g/m}^{3}$, respectively, and the entropy weight of O_3_ is in a downward trend in case two, it can be considered that the change in residents’ travel behavior will bring about an increase in O_3_ concentration. At the same time, the O_3_ concentration maintains a higher concentration level, from 0:00 to 24:00 in case two, instead of the peak value of O_3_ concentration, only from 13:00 to 19:00 PM, and a low concentration at other hours in case one. This should arouse public awareness of the development trend of O_3_ concentration after the change in residents’ travel behavior.

The maximum concentrations of CO, O_3_, PM_10_, PM_2.5_, NO_2_ and SO_2_ are 1190, 130, 73, 61, 57 and 20 ${\mu g/m}^{3}$, respectively, and their minimum concentrations in case two are 480, 4, 1, 1, 7 and 4 ${\mu g/m}^{3}$, respectively. The range (*R*) between the maximum and minimum concentration can be determined by Equation (19).
(19)$${R=P}_{max}{-P}_{min}$$
where, $P_{max}$ refers to the maximum concentration, $P_{min}$ means the minimum concentration.

The calculation results of Formula (19) show that the *R* of CO, O_3_, PM_10_, PM_2.5_, NO_2_ and SO_2_ are 710, 126, 72, 60, 50 and 16 ${\mu g/m}^{3}$, respectively. The Shannon entropy weight is the opposite, as the Shannon entropy weights of CO, O_3_, PM_10_, PM_2.5_, NO_2_ and SO_2_ in Table 4 are 0.083962, 0.126202, 0.132877, 0.145131, 0.216554 and 0.295273, respectively. This means that the greater the range (*R*) is, the smaller the entropy weight is. For this reason, when evaluating the quality of the atmospheric system in case two, it can be considered that CO contains less information, so it can be given less attention. In contrast, SO_2_ should be given more attention, because the abnormal discrete values of SO_2_ were found in Figure 7.

Nowadays, AQI is adopted to describe the extent of air pollution and the impact of air pollution on human health in China. It is calculated based on the concentration of six representative pollutants, including CO, SO_2_, NO_2_, O_3_, PM_2.5_ and PM_10_, and according to Equations (20) and (21), by converting the concentration of each representative pollutant into a comparable dimensionless individual air quality index (IAQI), and then taking the maximum IAQI to describe the situation of air pollution.
(20)$${\mathrm{IAQI}}_{P}=\frac{{\mathrm{IAQI}}_{Hi}{-\mathrm{IAQI}}_{\mathrm{Lo}}}{{\mathrm{BP}}_{\mathrm{Hi}}{-\mathrm{BP}}_{\mathrm{Lo}}}{(C}_{P}{-\mathrm{BP}}_{\mathrm{Lo}}{)+\mathrm{IAQI}}_{\mathrm{Lo}}$$
(21)$${\mathrm{AQI}=\max\{\mathrm{IAQI}}_{1}{,\;\mathrm{IAQI}}_{2}{,\ldots ,\mathrm{IAQI}}_{n}\}\;(n=1,\;2,\;\cdots ,\;P)$$
where, ${\mathrm{IAQI}}_{P}$ refers to the IAQI of pollutant *P*, and pollutant *P* can stand for SO_2_, CO or O_3_ or NO_2_ or PM_2.5_ or PM_10_. If ${\mathrm{IAQI}}_{P}$ > 50, then the pollutant *P* is defined as chief pollutant (CP). It is considered that the pollutant *P* does not affect human health if its ${\mathrm{IAQI}}_{P}$ ≤ 50. That is to say, the pollutant *P* will threaten human health if its ${\mathrm{IAQI}}_{P}$ > 50. If ${\mathrm{IAQI}}_{P}$ > 100, then the pollutant *P* is called excessive pollutant (EP). $C_{P}$ is the measured mass concentration of pollutant *P*. ${\mathrm{BP}}_{\mathrm{Hi}}$ indicates the breakpoint which is ≥$C_{P}$ in Table 5. ${\mathrm{BP}}_{\mathrm{Lo}}$ stands for the breakpoint which is ≤$C_{P}$ in Table 5. ${\mathrm{IAQI}}_{\mathrm{Hi}}$ indicates the IAQI corresponding to ${\mathrm{BP}}_{\mathrm{Hi}}$ in Table 5. ${\mathrm{IAQI}}_{\mathrm{Lo}}$ represents the IAQI corresponding to ${\mathrm{BP}}_{\mathrm{Lo}}$ in Table 5.

Within the total number of 547 observation times in case one, the CP appeared 392 times. Among them, the number of occurrences of one CP is 29, the number of simultaneous occurrences of two CPs is 180, the number of simultaneous occurrences of three CPs is 167, and the number of simultaneous occurrences of four CPs is 16. The occurrences of ${\mathrm{IAQI}}_{O_{3}}$, ${\mathrm{IAQI}}_{{\mathrm{NO}}_{2}}$, ${\mathrm{IAQI}}_{{\mathrm{PM}}_{2.5}}$, and ${\mathrm{IAQI}}_{{\mathrm{PM}}_{10}}$, which are greater than 50, are 14, 246, 156 and 271, respectively. The number of AQIs determined by ${\mathrm{IAQI}}_{O_{3}}$ is 14, the number of AQIs determined by ${\mathrm{IAQI}}_{{\mathrm{NO}}_{2}}$ is 453, the number of AQIs determined by ${\mathrm{IAQI}}_{{\mathrm{PM}}_{2.5}}$ is 18, and the number of AQIs determined by ${\mathrm{IAQI}}_{{\mathrm{PM}}_{10}}$ is 62. The EP occurred 32 times in total. Among them, the number of occurrences of one EP (all EP refer to NO_2_) is 28, and the number of simultaneous occurrences of two EPs (here, the EPs include NO_2_ and O_3_) is 4. The frequency of air quality being categorized into excellent, good and lightly pollution are 155, 360 and 32, respectively. In short, air quality is mainly determined by four pollutants, including NO_2_, O_3_, PM_2.5_, and PM_10_. At the same time, CP and EP were found before the changes in residents’ travel behavior. Within the total number of 547 observation times in case two, the frequency of air quality belonged to the excellent category, with no CP, and EP being found, is 457. The number of occurrences of one CP is 38, the number of simultaneous occurrences of two CPs is 52, the number of simultaneous occurrences of three or more CPs is 0. The occurrences of ${\mathrm{IAQI}}_{O_{3}}$ and ${\mathrm{IAQI}}_{{\mathrm{PM}}_{2.5}}$, which are greater than 50, are 8 and 82, respectively. EP was not found. The comparison between case one and case two reveals that the air quality has improved significantly, and the concentration of pollutants (including NO_2_, PM_2.5_, and PM_10_) that has posed a potential threat to human health decreased significantly after the changes in residents’ travel behavior. However, since the average hourly concentration of O_3_ has increased compared to that in case one, special attention should be paid to the development trend of O_3_.

Vehicular exhaust emissions have become the major source of urban air pollution, and the changing tendency of exhaust emissions is basically consistent with that of traffic flow [44]. In the first week that requires fewer travels from 24 January to 30 January 2020, the people do not need to go to work and school. Hence, the usual traffic is mainly changed to purchase daily necessities or meet other daily needs nearby. Additionally, the road traffic is no longer as busy as before. The pollutant concentrations of PM_2.5_, PM_10_, SO_2_, NO_2_, O_3_ and CO in this week are 17.94, 23.86, 4.88, 15.05, 49.92 ${\mu g/m}^{3}$ and 0.79 ${\mathrm{mg}/m}^{3}$, respectively. In addition, the average AQI is 33, and CP and EP are not found in this week. In the second week that requires fewer travels from 31 January to 6 February 2020, although the students have not yet started school, the enterprises’ production resumed, one after another. As a result, the amount of traffic on the road gradually increased. The pollutant concentrations of PM_2.5_, PM_10_, SO_2_, NO_2_, O_3_ and CO are 26.27, 33.25, 5.23, 21.25, 53.38 ${\mu g/m}^{3}$ and 0.69 ${\mathrm{mg}/m}^{3}$, respectively, and the average AQI is 41. Additionally, PM_2.5_ and O_3_ as the CPs were found in the second week. Similarly, In the third week, that requires fewer travels from 7 February to 13 February 2020, the pollutant concentrations of PM_2.5_, PM_10_, SO_2_, NO_2_, O_3_ and CO are 23.69, 31.14, 4.87, 22.46, 32.80 ${\mu g/m}^{3}$ and 0.76 ${\mathrm{mg}/m}^{3}$, respectively, and the average AQI is 38. PM_2.5_ as the CP was found in the third week. The pollutant concentrations of PM_2.5_, PM_10_, SO_2_, NO_2_, O_3_ and CO before the required fewer travels are 32.28, 61.38, 6.86, 52.22, 40.14 ${\mu g/m}^{3}$ and 0.98 ${\mathrm{mg}/m}^{3}$, respectively. Additionally, the average AQI before the travel reduction is 66. The comparison of these data including the data before the required fewer travels and the data of the first, the second as well as the third week that requires fewer travels, shows that the average concentration of pollutants is the lowest, and the air quality is optimal in the first week with fewer travels. This situation will become slightly worse in the second and third weeks with fewer travels. Therefore, it can be concluded that residents’ travel behavior is intrinsically linked to air quality. When the traffic volume on the road dropped sharply, the air quality improved significantly and the threat to human health was alleviated. 

## 5. Conclusions

In order to quantitatively analyze the relationship between abnormal human activities and air pollution from a new perspective, this study was conducted based on the real data from a case city in China. A series of detailed studies have been carried out using the correlation method, entropy weight method and AQI calculation method. Additionally, the research found that,

(1)The variation of residents’ travel behavior has brought down the concentrations of particulate matter. However, it has not changed the characteristics of the bimodal distribution of their concentrations. It will cause their concentration peaks to appear about an hour later in the morning and about an hour earlier in the evening. It is also one of the important factors in determining when the peak of SO_2_ concentration occurs. It will change the concentration distribution of O_3_ and NO_2_ from unimodal to bimodal. Additionally, it will cause a peak of NO_2_ concentration to appear two hours earlier in the evening.(2)The variation of residents’ travel behavior can promote a concentration reduction of particulate matter and gaseous contaminants. However, the dispersion of the hourly measured concentration of SO_2_, NO_2_, and PM_2.5_ has increased. Additionally, the dispersion of the other three pollutants including CO, PM_10_ and O_3_ is declining. Because SO_2_ and O_3_ have obvious discrete data values. Therefore, special attention should be given to gaseous contaminants, especially for the change in SO_2_ and O_3_ concentration, even if vehicles on the road are no longer as busy as usual and residents’ travel behavior is no longer the same.(3)The decline rates of the average concentrations of six representative pollutants in case two are significant. Among them, the decrease rate of NO_2_ and PM_10_ are the most considerable. The decrease rate of NO_2_ concentration is 61.05%, and the decrease rate of PM_10_ concentration is 53.68%. On the contrary, the concentration of O_3_ has increased significantly, and the growth rate of its average concentration has reached 9.82%.(4)Air quality is mainly determined by four pollutants, including NO_2_, O_3_, PM_2.5_, and PM_10_. At the same time, CP and EP were found before the changes in residents’ travel behavior. However, air quality has improved significantly, and the concentration of pollutants that have posed a potential threat to human health decreased significantly after the changes in residents’ travel behavior. The air quality is optimal in the first week, followed by the second and third weeks with fewer travels. Since the average hourly concentration of O_3_ has increased compared to that in case one, special attention should be paid to the development trend of O_3_.

These research results will have some guiding significance for the study of air pollution prevention, cleaner production, and indoor environmental safety issues, especially for the study of abnormal traffic environments where residents’ travel behaviors have changed significantly.

## Figures and Tables

**Figure 1 ijerph-17-04947-f001:**
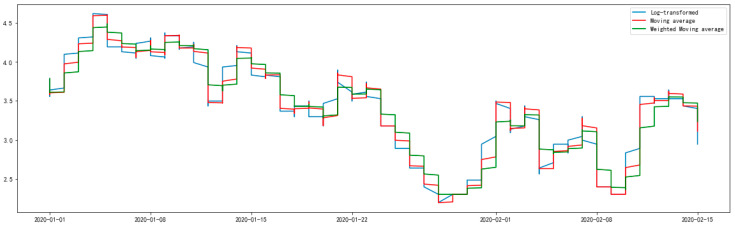
The log-transformed, moving average and weighted moving average of NO_2_ concentration datasets.

**Figure 2 ijerph-17-04947-f002:**
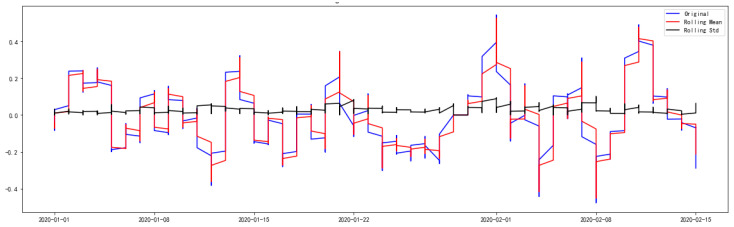
The rolling mean and rolling standard deviation of NO_2_ concentration datasets.

**Figure 3 ijerph-17-04947-f003:**
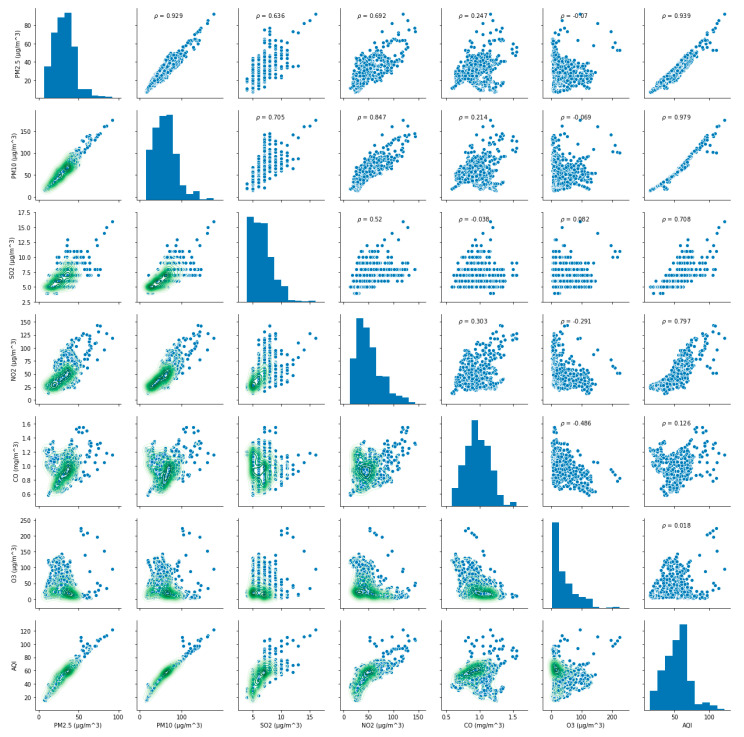
The correlation relationship between Air Quality Index (AQI) and six pollutants in case one.

**Figure 4 ijerph-17-04947-f004:**
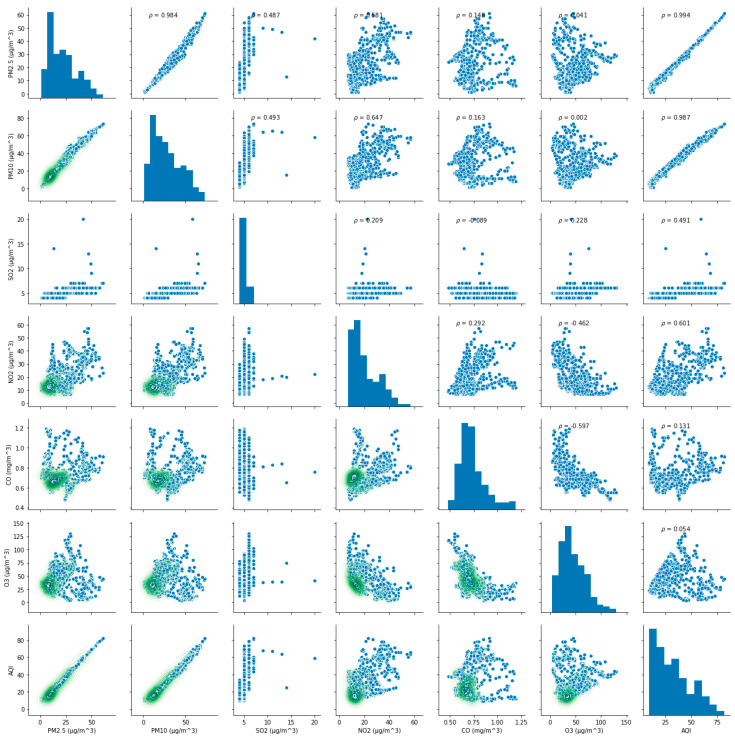
The correlation relationship between AQI and six pollutants in case two.

**Figure 5 ijerph-17-04947-f005:**
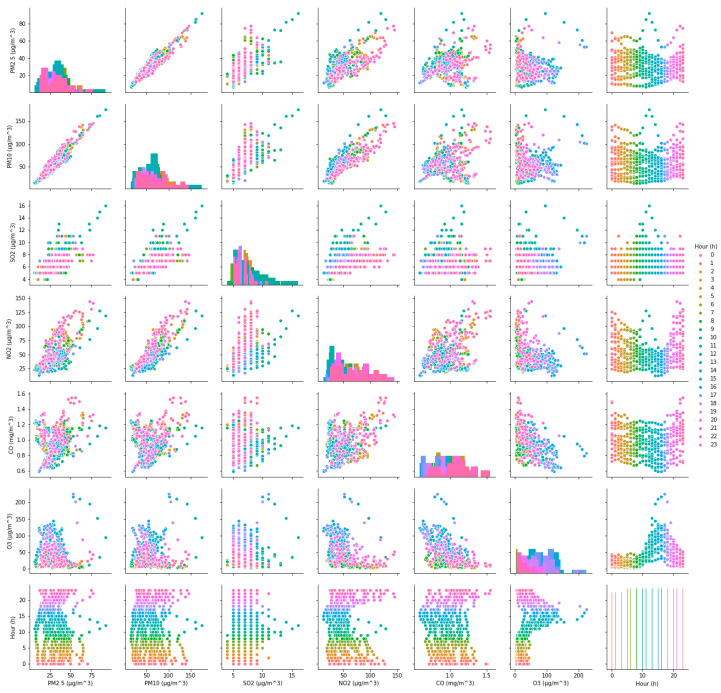
The pollutant concentrations distribution in case one.

**Figure 6 ijerph-17-04947-f006:**
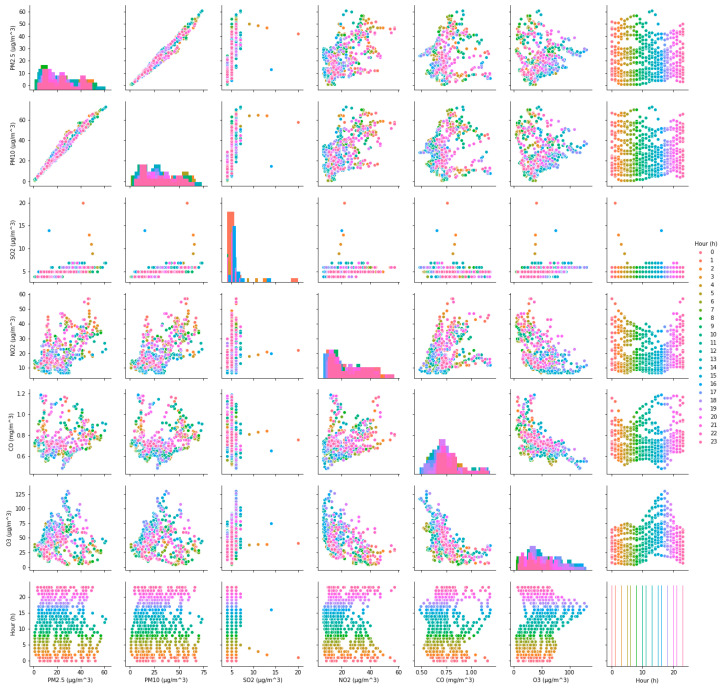
The pollutant concentrations distribution in case two.

**Figure 7 ijerph-17-04947-f007:**
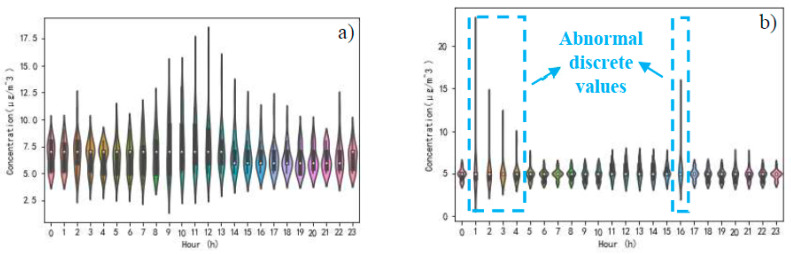
Time distribution characteristics of SO_2_ concentration, (**a**) in case one, (**b**) in case two.

**Figure 8 ijerph-17-04947-f008:**
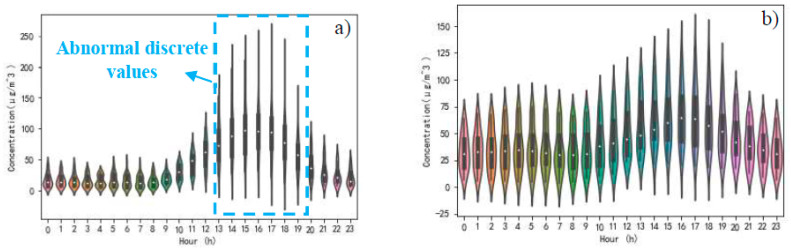
Time distribution characteristics of O_3_ concentration, (**a**) in case one, (**b**) in case two.

**Table 1 ijerph-17-04947-t001:** Results of augmented Dickey–Fuller (ADF) test before and after the data transformation of NO_2_.

	Before the Data Transformation	After the Data Transformation
Test Statistic	−1.273999	−5.592696
*p*-Value	0.640982	0.000001
Critical Value (1%)	−3.436442	−3.436364
Critical Value (5%)	−2.864230	−2.864195
Critical Value (10%)	−2.568202	−2.568184

**Table 2 ijerph-17-04947-t002:** The correlation coefficients between AQI and six pollutants.

	AQI	PM_2.5_	PM_10_	SO_2_	NO_2_	CO	O_3_
Coefficients							
Case one		0.939462	0.978872	0.707992	0.796998	0.125977	0.018185
Case two		0.994004	0.986916	0.491164	0.601048	0.130621	0.053570

**Table 3 ijerph-17-04947-t003:** Average concentration of six representative pollutants in two cases and their decline rates of average concentration in case two.

Average Concentration	PM_2.5_ (µg/m^3^)	PM_10_ (µg/m^3^)	SO_2_ (µg/m^3^)	NO_2_ (µg/m^3^)	CO (µg/m^3^)	O_3_ (µg/m^3^)
Case one	32.28	61.38	6.86	52.22	0.98	40.14
Case two	21.77	28.43	4.99	20.34	0.74	44.08
Decline rates	32.56%	53.68%	27.26%	61.05%	24.49%	−9.82%

**Table 4 ijerph-17-04947-t004:** Comparison of entropy weights of six pollutants.

	PM_2.5_	PM_10_	SO_2_	NO_2_	CO	O_3_
Case one	0.120497	0.145691	0.143843	0.157466	0.095655	0.336848
Case two	0.145131	0.132877	0.295273	0.216554	0.083962	0.126202
Weight adjustment	↑	↓	↑	↑	↓	↓

**Table 5 ijerph-17-04947-t005:** Individual air quality index (IAQI), concentration thresholds and air quality.

IAQI	Pollutant Concentration Thresholds						Air Quality Categories
	SO_2_ 24-h Average (µg/m^3^)	NO_2_ 24-h Average (µg/m^3^)	CO 24-h Average (mg/m^3^)	O_3_ 8-h Moving Average(µg/m^3^)	PM_2.5_ 24-h Average (µg/m^3^)	PM_10_ 24-h Average (µg/m^3^)	
0	0	0	0	0	0	0	Excellent
50	50	40	2	100	35	50	Excellent
100	150	80	4	160	75	150	Good
150	475	180	14	215	115	250	Light pollution
200	800	280	24	265	150	350	Moderate pollution
300	1600	565	36	800	250	420	Heavy pollution
400	2100	750	48	1000	350	500	Severe pollution
500	2620	940	60	1200	500	600	Severe pollution

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
