# Peer review of "Analysis on the Temporal Distribution Characteristics of Air Pollution and Its Impact on Human Health under the Noticeable Variation of Residents’ Travel Behavior: A Case of Guangzhou, China"

_ijerph, 2020, doi:10.3390/ijerph17144947_

Round 1
Reviewer 1 Report
I think the paper can be published in present form.
Reviewer 2 Report
I read the revised manuscript and the responses to the reviewers' comments, the authors replied to all the comments, modifying the manuscript accordingly and satisfactorily, or providing adequate answers to the doubts raised. I have no other comments and I believe that the manuscript can be accepted in this formThis manuscript is a resubmission of an earlier submission. The following is a list of the peer review reports and author responses from that submission.
Round 1
Reviewer 1 Report
Paper presents a challenging problem. The entropic measure of distance is also an interesting approach to measure dissimilarities .I suggest however do more test with other measures of distance or at least add a comment, why the selected one is better than other ones.
Since presented methodology is quite sensitive to departures from stationarity, I suggest also adding comments regarding stationarity tests of data set and necessary adjustments or transformations of the original data set in order to avoid nonstationarity.
Reviewer 2 Report
The subject presented in this paper is actual and sound appealing. The introduction is well constructed, but after that, the paper reveals a series of weaknesses. The authors used a set of data recorded in the Air Quality Monitoring Network Stations of Guangzhou Region (China Environmental Monitoring Station), where six parameters are measured, PM2.5, PM10, NO2, CO, O3 and SO2 and automatically recorded every hour. The authors are aware that the concentrations of pollutants measured at different stations are affected by many factors such as the geographic location of monitoring stations, the number of nearby residents, and the amount of road traffic. However they don’t distinguished the type of air quality station (hot spots, urban background, traffic) and used a pollutant hourly average parameter concentration of all stations to developed their analyze, which could introduce a bias in the data interpretation when it was not attempted. To add this weakness in the case one analyzed, before the breakdown in the urban traffic due to the COVID 19 pandemic, they use two weeks of data, whereas in case two, they use only one week. The sample size is quite different and for statistical and correlation purposes it must be evaluated. The authors could have overcome this gap by associating information on the reduction of emissions of primary pollutants (estimate) due to the drop in traffic intensity in the region and add to the discussion. The comparison of data between the two periods is done in global terms without considering a more detailed analysis of extreme cases that may have occurred in the two periods under analysis. Given the short period of data, the average concentration of each parameter ​​can be significantly affected. The authors in any part of the paper distinguished the source of pollutants and look likes that all are directly related to traffic. the authors note some trends in the behavior of variation in the daily concentration of pollutants but do not explore the factors that may be the cause of this behavior. They limit themselves to finding out. In the analysis of the behavior of ozone, for example, they do not assess the contribution of NOx emissions to ozone titration, which is markedly notorious for the night period. The higher average ozone concentrations in case two, may be mainly due to the contribution of the NOx emissions decrease (in exhaust emission NO>NO2) with less ozone consumption at night. If we look to the max ozone concentrations in the afternoon period, it seems that the recorded concentrations are similar, and reach higher hourly concentrations during the period of case 1. In relation to the information presented in fig. the authors forgot to identify units and most of the individual graphs in fig. 1 and 3 and 2 and 4 are repeated. In these figures they presented in diagonal histograms that are not identified what they want to show. The addition of entropy weight analyses to the data set it seems interesting, but the discussion still superficial. The paper try to answer to a challenge that we face presently due the impact of traffic in air quality but don´t add new scientific contribution to it.